# Directed Evolution of an Improved Rubisco; In Vitro Analyses to Decipher Fact from Fiction

**DOI:** 10.3390/ijms20205019

**Published:** 2019-10-10

**Authors:** Yu Zhou, Spencer Whitney

**Affiliations:** Australian Research Council Center of Excellence for Translational Photosynthesis, Research School of Biology, The Australian National University, 134 Linnaeus Way, Acton, ACT 0200, Australia; Yu.Zhou@anu.edu.au

**Keywords:** photosynthesis, carbon fixation, synthetic biology, metabolic engineering

## Abstract

Inaccuracies in biochemically characterizing the amount and CO_2_-fixing properties of the photosynthetic enzyme Ribulose-1,5-bisphosphate (RuBP) carboxylase/oxygenase continue to hamper an accurate evaluation of Rubisco mutants selected by directed evolution. Here, we outline an analytical pipeline for accurately quantifying Rubisco content and kinetics that averts the misinterpretation of directed evolution outcomes. Our study utilizes a new T7-promoter regulated Rubisco Dependent *Escherichia coli* (RDE3) screen to successfully select for the first *Rhodobacter sphaeroides* Rubisco (*Rs*Rubisco) mutant with improved CO_2_-fixing properties. The *Rs*Rubisco contains four amino acid substitutions in the large subunit (RbcL) and an improved carboxylation rate (*k_cat_^C^*, up 27%), carboxylation efficiency (*k_cat_^C^*/*K_m_* for CO_2_, increased 17%), unchanged CO_2_/O_2_ specificity and a 40% lower holoenzyme biogenesis capacity. Biochemical analysis of *Rs*Rubisco chimers coding one to three of the altered amino acids showed Lys-83-Gln and Arg-252-Leu substitutions (plant RbcL numbering) together, but not independently, impaired holoenzyme (L_8_S_8_) assembly. An N-terminal Val-11-Ile substitution did not affect *Rs*Rubisco catalysis or assembly, while a Tyr-345-Phe mutation alone conferred the improved kinetics without an effect on *Rs*Rubisco production. This study confirms the feasibility of improving Rubisco by directed evolution using an analytical pipeline that can identify false positives and reliably discriminate carboxylation enhancing amino acids changes from those influencing Rubisco biogenesis (solubility).

## 1. Introduction

Ribulose-1,5-bisphosphate carboxylase/oxygenase (Rubisco) is the initial, often rate-limiting, enzyme in the photosynthetic Calvin–Benson–Bassham (CBB) cycle that converts CO_2_ into the pre-cursor organic molecules that underpin life in the biosphere [1]. Rubisco constitutes 20–50% of the protein in leaves and 5–25% of the protein in algae and photosynthetic prokaryotes, making it the most abundant protein on earth [2,3,4]. This abundance is necessary due to the slowness of the rate at which Rubisco fixes CO_2_ to Ribulose-1,5-bisphosphate (RuBP) to produce duplicate 3-phosphoglycerate (3PGA) molecules—a reaction completing only 2–5 cycles per second in vascular plants. The carboxylation reaction rate is also compromised by the erroneous fixation of O_2_ instead of CO_2_, producing a 3PGA and 2-phosphoglycolate (2PG), a metabolically toxic product whose recycling back into 3PGA (from two 2PG) by photorespiration consumes energy and releases fixed CO_2_ [5,6]. In crops such as wheat and rice, 25% or more of the CO_2_ assimilated by photosynthesis is lost during photorespiration. Improving the carboxylation properties of plant Rubisco has therefore long been considered a valuable approach for enhancing crop photosynthesis and growth [7]. As more efficient Rubisco isoforms of an equivalent Form I structure exist naturally in some red algae, the feasibility of improving plant Rubisco appears tenable [8].

Form I Rubisco comprises a structurally conserved ~550 kD complex comprising a core of eight large (RbcL) subunits (arranged as a tetrad of RbcL_2_ dimers) capped by two tetrads of small (RbcS) subunits [9]. The eight catalytic sites in each L_8_S_8_ complex are formed by conserved amino acids in each adjoining RbcL2 pair (Figure A1). While we have a relatively detailed understanding of the conserved, complex catalytic chemistry of Rubisco [10,11] the rationale design of a better Rubisco remains elusive. Compounding the challenge is the finding that natural Rubisco kinetic diversity has evolved from amino acid differences outside of the catalytic pocket, sometimes in the distantly-located RbcS [12,13,14].

Defining what constitutes a better Rubisco is dependent on the cellular physiology of the target organism. In organisms incorporating a CO_2_-concentrating mechanism (CCM) to hinder Rubisco oxygenation (e.g., in most algae, cyanobacteria and plants with crassulacean acid metabolism, CAM, or C4-physiologies), improving the CO_2_ fixation rate (*k_cat_^C^*) would represent a better Rubisco [1]. In organisms lacking a CCM, such as C_3_ plants, the photosynthetic models show that improving Rubisco specificity for CO_2_ over O_2_ (S_C/O_) and increasing carboxylation efficiency (i.e., *k_cat_^C^* divided by the K_m_ for CO_2_ under ambient O_2_, *K_c_^21%O2^*) are key to producing a “better” Rubisco [15]. Rubisco from some red algae possesses these desired kinetic improvements. For example the S_C/O_ and *k_cat_^C^*/*K_c_^21%O2^* values for red algae *Griffithsia monilis* Rubisco are, respectively, two-fold and 1.3-fold higher than tobacco Rubisco [16]. While this kinetic improvement has the potential to improve plant growth by 30% [17], the assembly requirements of red algae Rubisco are unfortunately not met in tobacco chloroplasts [18,19]. A comparable molecular chaperone incompatibility also impairs, or prevents, the assembly of heterologous plant and cyanobacteria Rubisco in tobacco chloroplasts [20,21,22,23].

A successful approach to improving Rubisco kinetics is via the directed evolution of randomly mutated *rbcL* (+/− *rbcS*) libraries using high-throughput Rubisco-dependent *Escherichia coli* (RDE) screens [22,24,25]. As summarized in Figure 1, a core requirement shared among RDE screens is the expression of a *prk* gene coding phosphoribulose kinase (PRK). PRK catalyses the ATP-dependent phosphorylation of the ribulose 5-phosphate produced in the *E. coli* pentose phosphate pathway into RuBP. For unknown reasons, the PRK reaction is toxic to many *E. coli* strains unless Rubisco is introduced to catalyze the RuBP. As RDE selection is undertaken under elevated CO_2_, the RuBP is primarily carboxylated by Rubisco to produce the glycolytic intermediate 3PGA. Any 2PG produced via Rubisco oxygenase activity can also be metabolized by *E. coli* [26].

The life–death dependency of RDE screens on Rubisco activity has been exploited using the L-arabinose inducible P_BAD_ promoter to modulate PRK expression (i.e., RuBP production) to select for cells producing higher levels of Rubisco activity. This improvement to RDE fitness can stem from mutations in Rubisco that either improve its biogenesis (i.e., increasing the solubility of Rubisco) or/and increase its catalytic rate. Failure to accurately determine which of these Rubisco biochemical properties improves RDE selection fitness has led to solubility-enhancing RbcL substitutions being erroneously reported as mutations that enhance carboxylase activity (Table 1) [25].

The common Rubisco substrate used in directed evolution studies is that from the cyanobacterium *Synchococcus elongatus* PCC6301 (*Se*Rubisco, Table 1). When expressed in *E. coli*, the *Se*Rubisco RbcL and RbcS are produced in high abundance, with >90% of the RbcL produced forming misfolded insoluble aggregates, thereby limiting *Se*Rubisco expression to ~1% (*w*/*w*) of the soluble cellular protein [27,29,30,32]. Unassembled RbcS are less prone to insoluble aggregate formation, allowing them to rapidly bind (within <1 sec) to rbcL_8_ cores to form stable RbcL_8_RbcS_8_ (L_8_S_8_) complexes [33]. The limitations to GroELS chaperonin facilitating *Se*-RbcL folding and assembly in *E. coli* have led to the repeated selection of particular amino acids changes (e.g., RbcL residues 140, 189, 262, 345; Figure A1) that improve *Se*Rubisco biogenesis by >4-fold (Table 1). The impact of these mutations on catalysis is that they mostly impair—or at best provide only modest changes in—the carboxylation properties *Se*Rubisco.

A recent, two-tiered directed evolution study used *Te*-Rubisco from *Thermosynecococcus elongatus* whose assembly requirements are better met in *E. coli* (produced at ~6% (*w*/*w*) of the soluble cellular protein [22]). Using the “false-positive free” RDE2 screen (Figure 1), *Te*-Rubisco was evolved in sequential stages, with the first identifying the catalytically neutral P415A mutation that improved *Te*-Rubisco assembly 1.6-fold. The second cycle of evolution then successfully selected additional mutations to P415A that were all co-located at the RbcL–RbcS interface of the holoenzyme that could enhance *k_cat_^C^_,_ k_cat_^C^*/*K_c_^21%O2^* and *S_C/O_* (Table 1).

In this study, we trial a re-designed version of the RDE2 screen (here, named RDE3) to evolve Form I *Rs*Rubisco from *Rhodobacter sphaeroides*. Unlike *Se*Rubisco and plant Rubisco (that are both in the “green-type” Form IB lineage), *Rs*Rubisco is in the “red-type” Form IC lineage and more related to the catalytically superior Form ID algae Rubisco from *G. monilis* [34,35]. Unlike the RDE2 screen, *Rs*Rubisco production in the RDE3 system is regulated by bacteriophage T7 transcription and undertaken in JM109(DE3), a *lacUV5*-regulated T7 RNA polymerase-expressing *E. coli* strain sensitive to PRK expression. Described is a model-directed evolution experimental pipeline using RDE3 that distinguished amino acid substitutions that impair *Rs*Rubisco biogenesis from one that significantly enhanced *k_cat_^C^* and *k_cat_^C^*/*K_C_*.

## 2. Results

### 2.1. Ambiguity in How Cyanobacteria Rubisco Mutations Improve RDE Fitness

Prior RDE-directed evolution studies have mistakenly reported *Se*RbcL amino acid substitutions which improve the folding and assembly (solubility) of *Se*Rubisco as mutations that enhance catalysis. As summarized in Table 1, the commonly selected M262T *Se*RbcL substitution initially reported to enhance catalysis [36] was subsequently shown to enhance *Se*Rubisco solubility in *E. coli* [29] and may also account for the same mutation enhancing fitness in *R. capsulatus* selection [28]. Similarly the *Se*RbcL F140L and V189I mutations were initially characterized a catalytic enhancers of RDE fitness [27] but subsequently shown to improve *Se*Rubisco solubility by nearly 5-fold in *E. coli* [22].

The erroneous assessment of L_8_S_8_ recombinant holoenzyme production in *E. coli* often stems from the inherent complexity of the enzyme assembly requirements that impair its synthesis in heterologous hosts [24,25,37]. This is particularly true for the cyanobacteria Rubisco, whose assembly requirements are either not met, or are poorly met, in *E. coli* [29,32]. For example, the *Se*RbcL and *Se*RbcS subunits are produced in abundance in *E. coli*, with *Se*RbcL comprising >10% (*w*/*w*) of the total cellular protein (Figure 2a). Compared to *Se*RbcS that is mostly soluble, more than 90% of the *Se*RbcL produced in *E. coli* forms misassembled insoluble protein aggregates (compare the lysate and soluble *Se*RbcL and *Se*RbcS immunoblot signals in Figure 2b). The remaining soluble *Se*RbcL are primarily found assembled in *Se*L_8_S_8_ complexes, with the unassembled soluble *Se*RbcS monomers able to bind instantaneously with any assembled RbcL_8_ cores via a chaperone-independent process [33]. By comparison, the assembly requirements of *Rs*Rubisco are more readily met in *E. coli*, with a greater proportion of soluble *Rs*RbcL produced and assembled into *Rs*L_8_S_8_ complexes (Figure 2c).

The low level of *Se*Rubisco produced in *E. coli* (comprising ~1% (*w*/*w*) of the *E. coli*-soluble protein, Table 1) hinders the viability of SDS PAGE, Western blotting and native PAGE methods to quantify Rubisco content. Even when analyzed alongside known amounts of *Se*Rubisco, these methods can lack sufficient quantifiable resolution [38]. By contrast, the tight and stoichiometric binding properties of the Rubisco specific inhibitor carboxyarabinitol 1,5-bisphospahte (CABP) provide an accurate method to quantify Rubisco in any soluble cellular protein sample [39]. The method, termed ^14^C-CABP binding, takes advantage of CABP’s capacity to tightly bind to each Rubisco catalytic site to form a highly stable complex that, upon separation from unbound ^14^C-CABP (e.g., by gel filtration), provides an accurate quantitative measure of Rubisco content [38].

### 2.2. An Analytical Pathway for the Directed Evolution of Rubisco Using an RDE Screen

Our routine Rubisco-directed evolution experimental protocol spanning RDE selection to in vitro biochemistry analytics is summarized in Figure 3 [22,40]. The experimental pipeline begins with the generation and transformation of an *rbcL* (+/−*rbcS*) mutant library into an RDE screen alongside RDE cells expressing the non-mutated *rbcl–rbcs* gene control (Figure 3a). Colonies showing improved RDE fitness (i.e., those which can grow on PRK-inducing arabinose and IPTG-inducing Rubisco concentrations that have been pre-determined to be non-permissive to the growth of cells expressing the parental Rubisco substrate) are individually selected, and the *rbcl–rbcs* plasmid is isolated, retransformed into RDE and the colony growth on increasing arabinose compared with the control cells to gauge the relative improvement in cell “fitness” (Figure 3b). The *rbcl–rbcs* plasmid is isolated and sequenced from the RDE cells with improved fitness and transformed into a suitable *E. coli* strain to quantify changes in Rubisco expression and catalysis (Figure 3c). SDS PAGE samples of the total and soluble cell protein are taken to qualitatively assess the proportion of soluble and insoluble RbcL and RbcS produced (Figure 2). The soluble protein concentration is measured, the Rubisco content is quantified by ^14^C-CABP binding, and 4 to 8 μg of protein is separated by native PAGE to verify the L_8_S_8_ Rubisco content and aliquots used in ^14^CO_2_-fixation assays to quantify the CO_2_-fixation rate (*k_cat_^C^*) and the K_m_ for CO_2_ under anaerobic conditions (*Kc*), ambient O_2_ levels (*K_C_^21%O2^*), or over a range of O_2_ levels to quantify the K_m_ for O_2_ (*K_O_*) [22]. To accurately measure the specificity for CO_2_ over O_2_ (S_C/O_), the method of Kane et al. (1994) [41] is used, which requires Rubisco that has been purified from the *E. coli* using a rapid ion exchange and gel filtration process [42].

### 2.3. Determination of a Suitable E. coli Strain for RDE3

As in any directed protein evolution experiment, the level of success is dependent on the fidelity and throughput of the selection system employed. For evolving Rubisco, the use of the photosynthetic *R. capsulitis* bacterial screen is hampered by low throughput (screening is limited to a few thousand mutants) and hindered in its ability to characterize mutants with alterations in solubility [28,31,43,44]. As summarized in Figure 1, while RDE screens have higher throughput, their fidelity can be compromised by false positives. These occur at a higher frequency (~0.5% of the colony forming units plated) in the original RDE screens than in the metabolically altered MM1 RDE strain [24]. The false positives produced in MM1 selection have primarily been attributed to transposon insertions into a pACYC^RPK^ plasmid that silenced PRK production [25]. This has been avoided in the RDE2 screen by fusing NPTII onto the PRK C-terminus and including kanamycin in the selection media, meaning that false positives originating from transposon-induced PRK silencing cannot grow (Figure 1) [22].

In this study, we sought to adapt the RDE2 screen so that Rubisco production was regulated by bacteriophage T7 transcription rather than the previously used *lac* promoter in pTrcHisB. This necessitated that RDE selection be undertaken in a *lacUV5*-regulated T7 RNA polymerase-expressing *E. coli* strain. The plating of differing *E. coli* strains transformed with pACYC*^prk:kan^* on media containing differing arabinose concentrations found differences in their susceptibility to PRK-NPTII production (Figure 4). The JM109(DE3), XL1-Blue and DH5α strains were sensitive to PRK-NPTII synthesis, while DB3.1, BL21(DE3) and BW25113 were not, making the latter three strains unsuitable for use in an RDE screen. The pACYC*^prk:kan^* plasmids from DB3.1, BL21(DE3) and BW25113 colonies growing on 0.25% (*w*/*v*) arabinose were purified, restriction digested (Figure A2) and sequenced. There was no evidence that PRK-NPTII production was compromised, indicating that these three strains show an insensitivity to PRK production under the growth conditions trialed.

As JM109(DE3) is suited to T7-regulated recombinant expression, it was the strain used in our new RDE3 screen. Comparative expression analyses found that *Rs*Rubisco was expressed equally well in JM109(DE3) (~5% (*w*/*w*) of the soluble protein, see Table 2) as in BL21(DE3) (Figure 2).

### 2.4. Directed Evolution of RsRubisco in the RDE3 Screen

In this pilot RDE3-directed evolution experiment, the Form IC *Rs*Rubisco was the chosen substrate as (1) its assembly requirements are well met in *E. coli* (Figure 2) including in JM109(DE3) (Table 2), (2) it is of the “red-type” Rubisco phylogenetic lineage that encompasses the superior Form ID Rubisco isoforms from red algae [35], and (3) it has not been used in a directed evolution study. Electro-competent JM109(DE3) containing pACYC*^prk:kan^* [22] (here on termed RDE3 cells) were transformed with an *Rs–rbcl–rbcs* mutant library cloned into plasmid pYZ*Rs*LS (Amp^R^, Figure 5a). Approximately 10^5^ colony forming units (cfu) were plated on a 150 mm plate of selective LB media containing 75 μM IPTG, 0.25% (*w*/*v*) arabinose and antibiotics (Figure 5b). RDE3–pYZ*Rs*LS cells expressing non-mutated *Rs*Rubisco (2 × 10^4^ cfu; wild-type (WT) control) were grown on a 90 mm plate of replica selection media. After five days at 25 °C in air supplemented with 2% (*v*/*v*) CO_2_, 37 colonies had grown on the mutant *Rs*Rubisco plate and eight on the WT *Rs*Rubisco control (Figure 5b). The pYZ*Rs*LS from 33 mutant and all eight WT colonies were purified and restriction-digested to confirm they contained the *Rs–rbcLS* operon before re-transforming into fresh RDE3 cells and plating again onto LB containing 0.25% (*w*/*v*) arabinose (Figure 5c). Only one mutant, called YZ1, enhanced the growth fitness of RDE3 under this selection pressure, with the remainder showing comparable arabinose sensitivities to the pYZ*Rs*LS(WT) control (Figure 5c).

### 2.5. The RbcL Substitution Y345F Enhances RsRubisco Carboxylation without Impairing Enzyme Biogenesis

The sequencing of the YZ1 pYZ*Rs*LS plasmid found it coded five *Rs-rbcL* nucleotide mutations, with four coding the *Rs*RbcL mutations V11L (GTT-ATT), K85Q (AAA-CAA), R254L (CGT-CTT) and Y346F (TAT-TTT) and one a synonymous CTG-CTC change to L391. Relative to tobacco Rubisco, numbering these mutations corresponded to K83Q, R252L and Y345F (Figure A1a). A kinetic analysis of YZ1 *Rs*Rubisco found that the RbcL mutations had no effect on the *S_C/O_* but significantly stimulated *k_cat_^C^* and carboxylation efficiency (*k_cat_^C^* /*K_C_*) by 27% and 17%, respectively (Table 2). The mutations also increased the affinity of YZ1-Rubisco for O_2_ (i.e., *K_O_* was 25% lower than *Rs*Rubisco), thereby reducing its CO_2_ affinity under ambient O_2_ (*K_C_^21%O2^* = 78.8 μM) relative to wild-type *Rs*Rubisco (*K_C_^21%O2^* = 68.8 μM). Nevertheless, the carboxylation efficiency of YZ1-Rubisco in air (*k_cat_^C^/K_C_^21%O2^* = 59.6 mM^−1^·s^−1^) still exceeded *Rs*Rubisco (53.8 mM^−1^·s^−1^) by 11%.

The analysis of YZ1-*Rs*Rubisco solubility by ^14^C-CABP binding and native PAGE showed its biogenesis in JM109(DE3) was ~40% lower than wild-type *Rs*Rubisco (Table 2, Figure 6). A series of nine YZ1 *Rs*Rubisco chimers coding one to three RbcL amino acid substitutions was made (called YZ1a to YZi, Figure 5d) to examine whether the changes in biogenesis and kinetics could be uncoupled. As summarized in Table 2, all YZ1-chimers incorporating the Y345F mutation sustained the kinetics of YZ1 *Rs*Rubisco, demonstrating that this mutation imparted the improved *k_cat_^C^* and *k_cat_^C^*/*K_C_* kinetic phenotype. The assembly of *Rs*Rubisco was only impaired in the YZ1-chimers coding both K83Q and R252L together (YZ1, YZ1a, YZ1b, YZ1c, Table 2, Figure 6). On their own (YZ1f, Yz1g) or in combination (YZ1c), the K83Q and R252L substitutions had no significant influence on *Rs*Rubisco kinetics (Table 2). The separation of all the YZ1 *Rs*Rubisco chimers incorporating K83Q through native PAGE was marginally faster, suggesting that the mutation altered the net charge of the enzyme or induced a structural conformational change that increased its migration rate (Figure 6).

## 3. Discussion

A common challenge encountered with Rubisco-directed evolution studies using RDE selection is accurately measuring the biochemical properties of the mutant enzymes to correctly interpret the basis of how they improved RDE fitness [25]. Inaccuracies primarily arise from the erroneous evaluation of how amino acid substitutions affect Rubisco assembly (solubility) and kinetics (Table 1). Here, we demonstrate an experimental pipeline that spans the “selection” to the “analysis” phases of RDE-directed evolution (Figure 3). Using the pipeline, we successfully identified amino acids in Form IC “red-type” *Rs*Rubisco that influence either solubility or catalysis (Table 2). Our success in improving the carboxylation properties of *Rs*Rubisco extends to the comparable achievements made using MM1 and RDE2 in identifying amino acid substitutions that improve the carboxylase activities of the archaeon *Methanococcoides burtonii* Form III Rubisco decamer (rbcL_10_, [40]) and the Form IB “green-type” L_8_S_8_
*Te*Rubisco [22]. The chimeric dissection of the selected YZ1 *Rs*Rubisco mutant identified Y345F as a mutation that increases carboxylase activity and K84Q and R253L as substitutions that, in combination, impair L_8_S_8_ biogenesis (Table 2). Our findings also show that sensitivity to PRK production varies between *E. coli* strains (Figure 4) and that RDE false positive selection can arise independently from *prk* silencing (Figure 5b,c and Figure A2). This raises questions regarding the validity of the underpinning basis for RDE selection; i.e., that cell fitness is dependent on Rubisco expression to alleviate the inexplicable cellular toxicity of PRK-dependent RuBP production (Figure 1, [24]), a metabolic impediment that does not appear universal among all *E. coli* strains.

### 3.1. Mutating RbcL Amino Acid 345 Has Alternate Effects on Form I Rubisco Biogenesis and Catalysis

A common outcome of RDE screens is the selection of Rubisco mutants coding amino acid substitutions at residue 345 (numbered according to plant Rubisco, Table 1). Amino acid 345 is positioned in α-helix 6 of the six-stranded α/β-barrel C-domain of RbcL (as structurally defined by [45]). In the L_8_S_8_ structure residue, 345 is embedded beneath the enzymes’ outer surface, away from subunit interfaces and ~5.6 Å from Arg-295, a conserved catalytic site residue involved in binding the P5 phosphate group of RuBP (Figure 7). Adjoining α-helix 6 is the catalytic loop 6 (residues _331_AVGKLE_336_, Figure A1). This conserved, flexible, “lid-like” sequence interacts with the C-terminal tail to close over the catalytic pocket upon RuBP binding, then opening upon catalysis to allow product release [10]. Among Form IB and ID Rubisco isoforms, position 345 is phenylalanine (Phe, F), and in cyanobacteria Form IA Rubisco and *Rs*Rubisco Form IC, it is a tyrosine (Tyr, Y) (Figure A1). A mutagenic analysis of Form IB *Se*Rubisco found changes to α-helix 6 amino acids located closer to loop 6 dramatically impaired carboxylase activity [46]. Not included in this analysis were changes to the other α-helix 6 G^345^FVDL residues that are conserved among Form IB Rubisco. From this analysis, it was hypothesized that structural changes in α-helix 6 can influence loop 6 positioning and movement [46]. Such movements are thought to alter how K334 (the catalytically critical loop 6 apex residue) ligands with the RuBP enediolate transition state intermediate and influences its catalysis with CO_2_ or O_2_.

The existing data suggest that changes to residue 345 have a pervasive influence on Rubisco catalysis. For example, replacing Phe 345 with Ile, Leu or Val impairs the carboxylation properties of *Se*Rubisco [27,30] and *Te*Rubisco [22] (Table 1). It is difficult to structurally ascertain whether these kinetic changes arise from the mutations altering the positioning and movement of loop 6 or by altering RuBP ligand binding with Arg-295 in the catalytic pocket. Inexplicably, mutations to F345 also somehow impart two to 11-fold improvements in *Te*Rubisco and *Se*Rubisco production (Table 1). It is presumed that these changes enhance *Se*RbcL and *Te*RbcL folding and/or assembly in *E. coli* [22,24,27]. By contrast, the Y345F substitution in *Rs*RbcL had no effect on *Rs*Rubisco biogenesis in *E. coli* and improved its carboxylation rate and efficiency by 27 and 17%, respectively (Table 2). This incongruence in how changes to amino acid 345 differentially influence the catalysis and assembly of Form IB and IC Rubiscos underpins the challenge of using rationale design approaches to identify “all-purpose” amino acid substitutions that can benefit catalysis between, and within, differing Rubisco lineages [1,13,22,47].

### 3.2. How Do the K83Q andR252L Substitutions Function Cooperativly to Reduce RsRubisco Biogenesis?

K83 and R252 in *Rs*Rubisco are located, respectively, in strand C of the RbcL N-domain and α-helix 3 of the α/β-barrel (Figure 7a). Their distant locations from the catalytic sites are in accordance with our findings that the mutations K83Q and R252L, both singly (*Rs*Rubisco chimers YZ1f and YZ1g) and in combination (YZ1c), have no significant effect on catalysis (Table 2). *Rs*Rubisco biogenesis was not affected in YZ1 chimers containing only one of these mutations (chimers YZ1d to g), but was reduced by ~40% when both were present (YZ1 and YZ1a to c, Table 2). How the K83Q and R252L substitutions work in combination, but not in isolation, to reduce the level of *Rs*Rubisco is unclear. It is possible that the mutations reduce *Rs*RbcL compatibility with one or more components of the *E. coli* protein folding machinery that impairs holoenzyme assembly or alters the stability of the L_8_S_8_ complex. As summarized in Table 1, a number of single amino acid mutations are known to enhance cyanobacteria Rubisco biogenesis in *E. coli*. For *Te*Rubisco, the solubility-enhancing F345I and P415A mutations were able to function in a cumulative fashion to further augment solubility (Table 1). It is proposed that these improvements arose by improving the compatibility of RbcL with components of the *E. coli* protein folding machinery [22]. It is known that chaperonin (GroELS) availability, but not DnaK, DnaJ, GrpE or Trigger Factor, has a pervasive influence on *Se*Rubisco biogenesis in *E. coli* [29,32]. Sequence complementarity with *E. coli* GroELS has been used to explain the vast difference is the assembly capabilities of differing cyanobacteria Rubisco (e.g., PCC7002 Rubisco is produced at <0.5% CSP [32], *Se*Rubisco at ~1.3% CSP and *Te*Rubisco at ~7% CSP [22,48]) and with *Rs*Rubisco (5% CSP, Table 2). It is therefore possible that the K84Q and R253L mutations in combination reduce *Rs*RbcL compatibility with GroELS to impair *Rs*Rubisco biogenesis.

The evaluation of the *Rs*Rubisco L_8_S_8_ crystal structure (Figure 7b) showed that R252 is located near the surface of the inner solvent channel that traverses the L_8_S_8_ structure and is situated at the edge of the interface of adjoining large subunits that are arranged head to tail in each L_2_ unit (Figure 7c). The adjacent R253 residue is conserved among Form I Rubisco and forms a conserved salt bridge with E109 in the adjoined RbcL of each L_2_ (Figure 7c). While this intradimer interaction is hypothesized to maintain L_2′_s structural integrity [49], our findings suggest the R252L substitution does not disturb this Arg–Glu salt bridge as *Rs*Rubisco synthesis and stability seems unaffected (Table 2). In contrast to R252, K83 is a solvent-exposed outer surface residue and independent of any inter-subunit interactions (Figure 7b). Curiously, the YZ1 chimers incorporating K83Q showed a faster mobility through native PAGE, independent of other amino acid substitutions (Figure 6), suggesting the mutation alters the net surface charge of the protein or influences *Rs*Rubisco quaternary structure, but without an effect on catalysis (Table 2). The disparate locations of the K83R and R252L mutations in the *Rs*Rubisco do not support the hypothesis that they jointly function to reduce *Rs*Rubisco content in *E. coli* through destabilization of the L_8_S_8_ complex.

### 3.3. Understanding the Basis of RDE Selection

The adage that “you get what you screen for” highlights how crucial it is to ensure any directed evolution screen can select for a desired fitness trait with high fidelity [50,51]. The production of false positives commonly compromises the fidelity of RDE screens, especially those using PRK expressing wild-type *E. coli* (Figure 1a). In the MM1 *gapA*^−^ mutant RDE screen, the impaired cell viability associated with blocking glycolysis/gluconeogenesis by deleting glceraldehyde-3-phosphate dehydrogenase (Figure 1b) reduces the frequency of false positives [24,26] which primarily occur from the transposon-interrupted expression of PRK [25]. The selection of these false positives in the XL1-Blue-based RDE2 screen was prevented using PRK:NPTII fusion so that PRK production and kanamycin resistance are linked, thereby preventing the selection of PRK-silenced mutants (Figure 1c). While initial tests indicated the suitability of the JM109(DE3) strain for RDE3 selection (Figure 4), a false positive production rate of approximately 0.04% was found when more cells were grown on non-permissive, high arabinose LB media (Figure 5b). Curiously, no sequence changes were evident in the purified pACYC*^prk:kan^* or the pYZ*Rs*LS from the eight WT control colonies that grew. A comparable frequency of false positives was also identified in the RDE3-mutant *Rs*Rubisco screen (Figure 5b), in which again the sequence integrity of pACYC*^prk:kan^* was maintained, and only one (YZ1) of the 33 mutant pYZ*Rs*LS plasmids re-transformed into RDE3 survived re-selection on high arabinose. The growth of the false positives in the initial RDE3 screens therefore arose independently of changes to either RPK:NPTII or *Rs*Rubisco expression. This suggests the false positives may result from chromosomal mutations that alter JM109(DE3) cellular metabolism to dampen its sensitivity to PRK production. This hypothesis is consistent with our finding that other *E. coli* strains (e.g., DB3.1, BL21(DE3) and BW25113) show resistance to PRK:NPTII production (Figure 4). Current research is underway to identify the genetic and/or metabolic basis for the contrasting sensitivities of *E. coli* strains to PRK production as a means to improve the fidelity of RDE screens.

### 3.4. Pathways for Evolving RsRubisco Further—Why and How?

Our interest in continuing to explore the evolutionary potential of *Rs*Rubisco stems from its relatedness to the Form ID “red-Rubisco” lineage that comprises red algae Rubisco isoforms whose high *S_C/O_* and carboxylation efficiencies dramatically exceed those of crop plants [16,18]. While optimizing the fidelity of the RDE3 to avoid false positive selection is desirable, we found it is not critical if suitable care is taken to re-screen selected *Rs*Rubisco mutants (i.e., confirming that Rubisco is improving RDE3 fitness, Figure 5c) before pursuing Rubisco biochemical analyses. As the carboxylase improved YZ1i (Y345F) and biogenesis-impaired YZ1c (R83Q, R252L) mutants offer alternative starting points in the *Rs*Rubisco evolutionary landscape, separately evolving them using RDE3 and comparing the biochemical fitness traits selected (solubility and/or kinetic) would provide valuable strategical guidance to successive rounds of evolution.

Incorporating the *R. sphaeroides* chaperonin folding machinery components into RDE3 would alter—and possibly benefit—the *Rs*Rubisco evolutionary fitness landscape that can be surveyed. A similar approach taken with *Se*Rubisco found that the inclusion of the Rubisco-specific post-chaperonin assembly chaperone RbcX altered the fitness landscape of the MM1 screen, with different mutants selected compared with a screen lacking RbcX [27]. This supported an earlier hypothesis that Rubisco evolution is influenced—and possibly even constrained—by the need to maintain complementarity with the protein folding/assembly machinery of a cell [24]. This appears particularly apparent for Form IB plant Rubiscos, whose assembly in *E. coli* requires five compatible molecular partners (i.e., the Rubisco-specific chaperones Raf1, RbcX, Raf2, BSD2 and Cpn60αβ20 chaperonin cages [37,52,53]). Thus far, maintaining sequence compatibility between Raf1 and RbcL appears to be the most important factor influencing plant Rubisco biogenesis [54]. Our findings that the R83Q and R252L substitutions inexplicably impair *Rs*Rubisco biogenesis in *E. coli* suggest sequence compatibility between RbcL and that the bacterial GroELS protein folding machinery also impacts Rubisco biogenesis potential.

## 4. Materials and Methods 

### 4.1. Comparing the PRK Sensitivity of Different E. coli Strains

The pACYC*^prk:kan^* plasmid [22] was transformed into CaCl_2_-competent JM109(DE3), XL1-B, DH5α, DB3.1, BL21(DE3), and BW25113 *E. coli* cells, and a colony of each was grown in LB-Chlor (30 μg·mL^−1^) at 25 °C. At an absorbance (A_600_) of 0.6, the cultures were diluted by 10^−4^ and 20 µL was plated onto LB-Chlor plates containing 0, 0.05%, 0.1%, 0.25% (*w*/*v*) l-arabinose with/without 100 μg·mL^−1^ kanamycin. Colony growth at 25 °C in air containing 2% (*v*/*v*) CO_2_ was monitored up to 5 days. Electro-competent RDE3 cells (i.e., pACYC*^prk:kan^* transformed JM109(DE3)) in 10% (*v*/*v*) glycerol were N_2_ frozen and stored at −80 °C.

### 4.2. Library Construction and Rubisco Selection Using the RDE Screening 

Figure 3 provides a general summary of an RDE-directed evolution experiment. The pilot RDE3 *Rs*Rubisco evolution experiment undertaken in this study is summarized in Figure 5. The 1965-bp synthetic *Rs-rbcLS* operon in which the codon use was modified to match the tobacco *rbcL* gene (Genbank Accession KM464722) was PCR-amplified using primer 1355 (5′-ATCGAGGTCTCGCCATATGGCACCACAAACAGAGAC-3′, *BsaI* site underlined) and primer 1356 (5′-GCATGGTCTCCAAGCATCTCGAGCTCAGATCTGTC-3′, *BsaI* site underlined) and cloned into a modified pTriEx-2 Amp plasmid (Novagen, Addgene, Cambridge, Massachusetts, USA) to give plasmid pYZRsLS (Genbank Accession MN541175, Figure 5a). The 13 N-terminal amino acids of *Rs*RbcL coded in pYZRsLS have been replaced with the 12 N-terminal amino acids from the tobacco RbcL (Figure A1). In pYZRsLS the *BsaI* sites flanking the *Rs-rbcLS* operon are positioned downstream of the T7 promoter and adjacent to the T7 terminator. A mutant *Rs-rbcLS* library was amplified from pYZRsLS with primers 1355 and 1356 using the Genemorph II Random Mutagenesis kit (Agilent Technologies, Santa Clara, CA, USA) over 25 cycles as per the manufacturer’s recommendations. The amplified *Rs-rbcLS* mutant library was digested with *BsaI*, ligated into *BsaI* cut pYZRsLS and electroporated (1.8 kV pulse) into 25 μL RDE3 cells. RDE3 was also transformed with non-mutated pYZRsLS as the wild-type (WT) control. The cells were added to 1 mL LB, grown for 1 h at 37 °C, 5 μL diluted and plated on non-selective media (LB-chlor-amp; 16 µg/mL chloramphenicol and 100 µg/mL ampicillin) to determine the colony forming unit (cfu) size of the WT and mutant *Rs*Rubisco library. The mutant pYZRsLS plasmids from 10 colonies grown on permissive media were transformed into XL1-Blue (LB-amp selected), purified and fully sequenced. The *Rs-rbcLS* mutational frequency was 2.4 mutations per kb.

The remaining transformed RDE cells were pelleted (6000× *g*, 1 min), suspended in 0.2 mL LB and plated onto LB-arabinose selection (LB-chlor-amp plates containing 125 µg/mL kanamycin, 75 µM IPTG and 0.25% (*w*/*v*) arabinose, Figure 5b). The RDE3-WT and RDE3-mutant *Rs*Rubisco library plates were grown at 25 °C for 3–7 days in air containing 2% (*v*/*v*) CO_2_. All colonies from both plates were colony picked and grown in 1 mL LB-chlor-amp, their plasmids were purified and the pYZRsLS re-transformed into XL-1Blue (LB-Amp selection), and then purified for *NdeI* mapping (expect 3822, 2177 and 1020-bp products) before transforming into RDE3 for the re-screening of selection fitness.

Colonies of the RDE3-rescreened cells were grown to an OD_600_ of 0.6 in 1 mL LB-chlor-amp, diluted 10^−4^ with LB and 20 μL of cells plated on LB-arabinose selection (Figure 5c). Of the 41 plasmids re-screened (including eight from the WT screen, Figure 5b) only one, the pYZ1 mutant, improved RDE3 growth on the LB-arabinose (0.25% *w*/*v*, Figure 5c). The *Rs-rbcLS* operon in pYZ1 contained five nucleotide substitutions that coded four RbcL amino acid changes, from which nine YZ1 *Rs*Rubisco derivatives were generated by restriction cloning so that each chimer only coded one to three of each mutation (Figure 5d).

### 4.3. Rubisco Content and Catalysis

JM109(DE3) transformed with pYZRsLS and each pYZ1 derivative were grown at 28 °C and *Rs*Rubisco expression induced for 6 h with 0.5 mM IPTG before harvesting by centrifugation (10 min, 4 °C, 6200× *g*) and replica cell pellet aliquots N_2_ frozen and stored at −80 °C (Figure 3c). The soluble and total (lysate) proteins in cell pellets from 10 mL of cultured cells were analyzed by SDS-PAGE, native PAGE and immunoblot analysis as described [55]. The Rubisco content in the *E. coli* soluble protein was quantified by ^14^C-CABP binding as described [38]. The same soluble protein sample was used to measure *k_cat_^C^* and *K_C_* using ^14^CO_2_-fixation assays as described [22] using 0–170 μM ^14^CO_2_ in assays equilibrated with 0, 20%, 35% or 50% O_2_ (*v*/*v*) in N_2_. The increase in *K_C_* with increasing [O_2_–] was plotted and the slope of the linear fit used to derive *K_O_*. *S_C/O_* was quantified using the [1-^3^H]-RuBP consumption assay as described [41] using *Rs*Rubisco purified from ~40 mL of cultured cells (Figure 3c) by rapid ion exchange and size exclusion (Superdex 200, GE Life Science, Chicago, IL, USA) as described [42]. Soluble protein concentration was measured using the dye-binding Coomassie Plus Kit (Thermo Fisher Scientific, Waltham, MA, USA) against bovine serum albumin.

## 5. Conclusions

This study provides another demonstration of how directed evolution can improve the carboxylase properties of Rubisco. As shown here, key challenges with RDE screens include differentiating the Rubisco mutants that enhance cell fitness from false positives and then distinguishing carboxylation enhancing mutations from those that improve Rubisco biogenesis. Success is also dependent on tailoring selection screen conditions to the substrate Rubisco being evolved; that is, determining the optimal arabinose and IPTG concentrations for PRK and Rubisco production, keeping in mind that Rubisco production itself impairs *E. coli* viability [25]. An ongoing consideration is the extent to which the red-type *Rs*Rubisco might serve as a suitable surrogate to sequentially evolve its properties towards the high *S_CO_*, high carboxylation efficiency properties of the related “better” red algae Rubisco. One possible objective might be to replace plant Rubisco with a kinetically improved *Rs*Rubisco, although this will depend on whether its folding and assembly requirements are met by plant chloroplasts. Current evidence suggests the extent of prokaryotic Rubisco expression in *E. coli* is a poor indicator of its biogenesis potential in tobacco [20,21,22]. An alternative to evolving *Rs*Rubisco might be to evolve plant Rubisco itself using an RDE screen incorporating the chloroplast protein folding machinery [53,56]. As the components of the plant Rubisco expression systems are primarily T7-promoter regulated [37,57], our JM109(DE3)-based RDE3 screen appears to be a suitable selection host for such a directed evolution endeavor.

## Figures and Tables

**Figure 1 ijms-20-05019-f001:**
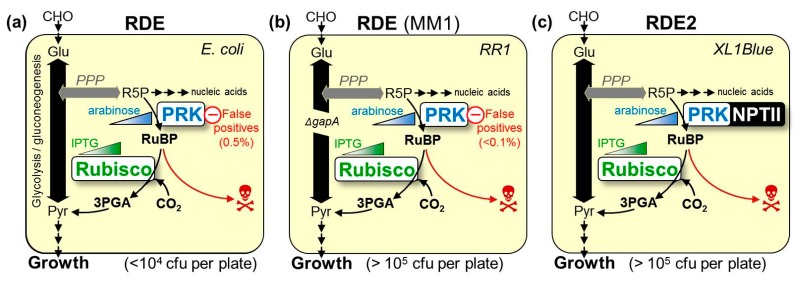
Metabolic rewiring in the different Rubisco dependent *E. coli* (RDE) screens. All RDE screens contain arabinose inducible BAD promoter-regulated *prk* genes whose product, phosphoribulokinase (PRK), phosphorylates ribulose 5-phosphate (R5P, produced in the pentose phosphate pathway, *PPP*) into RuBP. The toxicity of this reaction can be alleviated by expressing Rubisco to catalyse RuBP carboxylation into metabolically compatible 3-phosphoglycerate (3PGA). Cell growth can also arise from mutations that silence PRK to produce false positives (in which colony growth is independent of Rubisco activity). (**a**) The frequency of false positives in wild-type *E. coli* RDE screens is high (~0.5% of plated cells, left panel) reducing the number of colony-forming units (cfu) that can be effectively screened per plate. (**b**) The false positives frequency is >5-fold lower in the MM1 RDE system (middle panel) in which the *gapA* gene is deleted to stop flux through the glycolysis pathway [24]. (**c**) In RDE2, expressing PRK as an neomycin phoshotransferase II (NPTII) fusion prevents false positive selection, as silencing PRK co-suppresses NPTII expression, resulting in kanamycin sensitivity [22].

**Figure 2 ijms-20-05019-f002:**
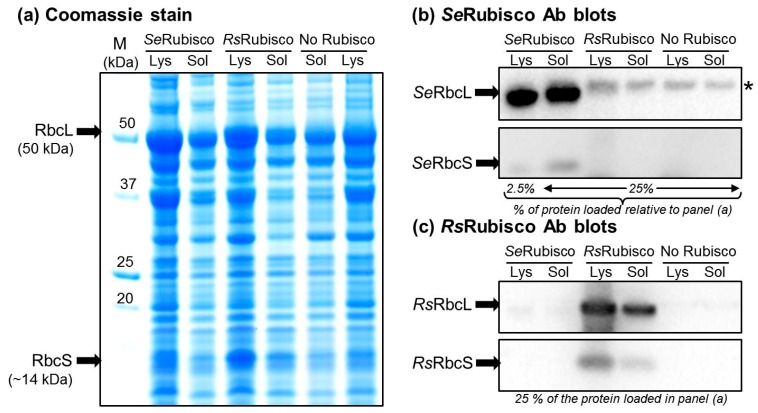
SDS PAGE analysis of *Se*Rubisco and *Rs*Rubisco production in BL21(DE3). Rubisco production was induced at 25 °C with 0.5 mM isopropyl-β-d-thiogalactosidase (IPTG) for 8 h, the cells were lysed and (**a**) 4 μg of soluble protein (Sol) and an equivalent volume of the total cellular protein (Lys, comprising soluble, membrane and insoluble inclusion body proteins) were separated by SDS PAGE and Coomassie stained. Duplicate PAGE gels loaded with lower amounts of each protein preparation (as indicated) were blotted onto nitrocellulose membranes and probed with antibodies (Ab) to (**b**) *Se*Rubisco and (**c**) *Rs*Rubisco. Empty vector BL21(DE3)-pET28a(+) cells were used as a “no Rubisco” control. *, *E. coli* protein recognized by the *Se*Rubisco antibody. The *Se*RbcL is produced in high abundance and mostly insoluble (lysate signal >> soluble signal), while *Se*RbcS, *Rs*RbcL and *Rs*RbcS are mostly soluble. *Se*Rubisco and *Rs*Rubisco L_8_S_8_ contents comprised, respectively, 1.1% and 4.8% (*w*/*w*) of the soluble cellular protein (as quantified by ^14^C- carboxyarabinitol 1,5-bisphospahte (CABP)-binding). M, protein markers (sizes shown).

**Figure 3 ijms-20-05019-f003:**
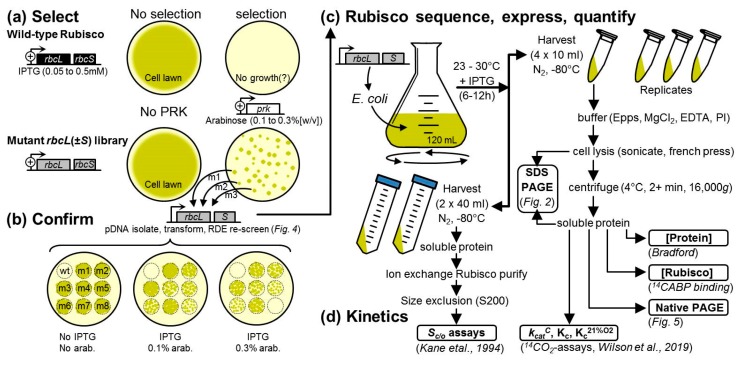
The analytical pipeline of an RDE experiment. An RDE-directed evolution project begins with (**a**) Rubisco gene library generation, transformation and RDE selection. The Rubisco gene plasmid (*rbcl–rbcs*) from cells with improved fitness is (**b**) transformed into fresh RDE cells and re-screened on PRK-inducing arabinose to confirm the improved growth phenotype before (**c**) sequencing to ascertain the amino acid substitutions. Each Rubisco mutant is then expressed in E. coli and the Rubisco subunit folding/assembly capacity evaluated by SDS PAGE (e.g., Figure 2); the L_8_S_8_ holoenzyme content is then quantified (^14^C-CABP binding), confirmed (native PAGE) and (**d**) the kinetics (S_C/O_, k_cat_^C^, K_C_ and K_C_^21%O2^) measured as described in [22].

**Figure 4 ijms-20-05019-f004:**
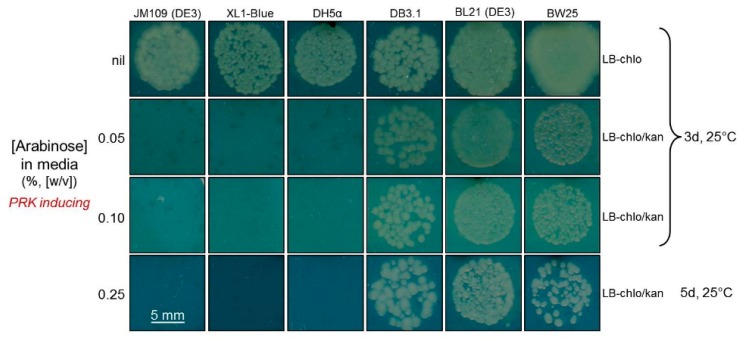
Comparative growth of differing *E. coli* strains under increasing PRK-NPTII expression. Six *E. coli* strains were transformed with pACYC*^prk:kan^* and grown on LB-medium containing four different arabinose concentrations. Chloramphenicol (chlo, 30 μg/mL^−1^) and kanamycin (kan, 125 μg/mL^−1^) were included in the LB plates, as indicated.

**Figure 5 ijms-20-05019-f005:**
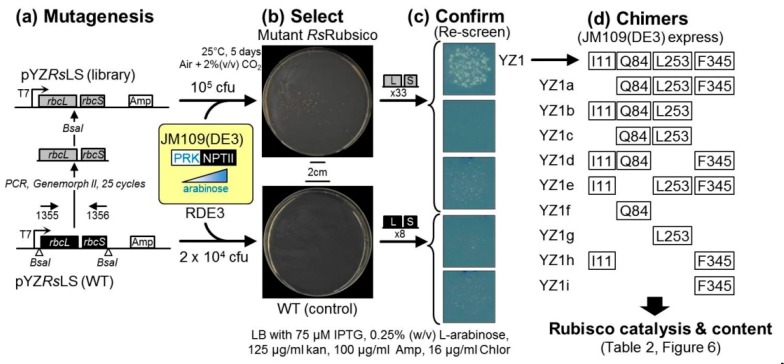
RDE3 evolution of *Rs*Rubisco. (**a**) Wild-type (WT) and a mutated *Rs–rbcLS* operon library cloned into pYZ*Rs*LS were transformed into RDE3 and (**b**) screened under strong arabinose selection as shown. The surviving colonies were grown (under non-selective conditions), the pYZ*Rs*LS purified and those containing the *Rs–rbcLS* operon re-transformed into RDE3 and (**c**) re-grown on selection media (examples for five of the 41 RDE3 re-screens are shown). Only the YZ1 mutant improved RDE3 growth, and (**d**) the sequencing of its *Rs–rbcLS* operon showed that it coded RbcL mutations V11L, K84Q, R253L and Y345F (plant RbcL numbering, Table A1). Nine YZ1 chimers (named YZ1a to YZ1i) coding differing combinations of these mutations were created to study how they influence *Rs*Rubisco biogenesis (Figure 6) and catalysis (Table 2).

**Figure 6 ijms-20-05019-f006:**
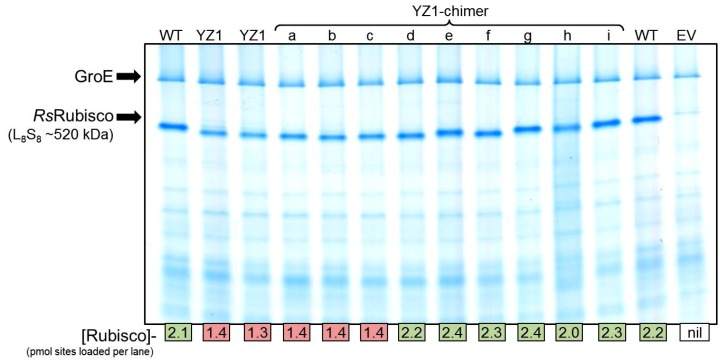
Native PAGE analysis of *Rs*Rubisco and the YZ1 chimers’ relative expression in JM109(DE3) after 0.5 mM IPTG induction for 6 h at 28 °C. The separated soluble cellular protein (4 μg, including an empty vector (EV) JM109(DE3)-pET28a(+) control) was Coomassie stained, with the *Rs*Rubisco band intensity correlating with that quantified by ^14^C-CABP binding (indicated at base of gel). *Rs*Rubisco biogenesis for the YZ1 chimers similar to wild-type (WT) are shaded green, while those produced in amounts equivalent to YZ1-*Rs*Rubisco are shaded red.

**Figure 7 ijms-20-05019-f007:**
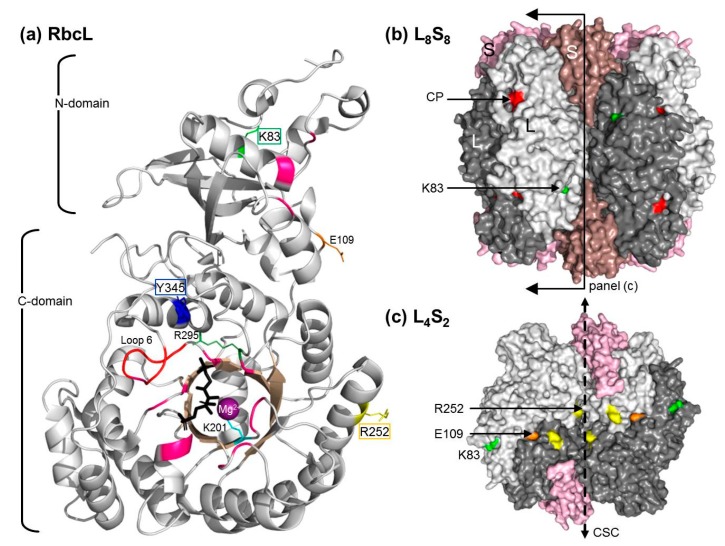
Mapping the location of the RDE3-selected YZ1 mutations in *R. sphaeroides* Rubisco. Pymol-assembled *Rs*Rubisco (PDB 5NV3) structures of (**a**) a single *Rs*RbcL subunit, (**b**) L_8_S_8_ holoenzyme (side view) and (**c**) L_4_S_2_ sub-complex (cross-section view). RbcL pairs are colored with differing shades of grey, and RbcS is alternately colored pink/salmon. Shown are the locations of the YZ1-selected RbcL amino acid substitutions K83Q (green, strand C), R252L (yellow, α-helix 3) and Y345F (blue, α-helix 6) relative to the conserved catalytic pocket (CP) residues (dark pink), including R295 (green), K201 (light blue) loop 6 (red) and CABP (black, RuBP intermediary analog). E109 (orange) forms a salt bridge with a conserved R253 in the adjacent RbcL. The R252L substitution is located on the surface of the central solvent channel (CSC) that traverses the inner core of the L_8_S_8_ holoenzyme (dashed arrow, panel c). Residues are numbered relative to plant Rubisco using the alignment in Figure A1.

**Table 1 ijms-20-05019-t001:** Influence of commonly selected cyanobacteria Rubisco mutants on Rubisco content and catalysis (relative to wild-type, WT)—deciphering the trait that improves fitness.

Amino Acid Mutation		*^a^*Trait Selected	Rubisco (Solubility)	*k_cat_^C^*	*K_C_*	*K_C_^air^*	*k_cat_^C^/K_C_*	*k_cat_^C^/K_C_^air^*	*S_C/O_*	^b^ Selection Host	Reference
***Se*Rubisco (*Synechococcus* sp. PCC 6301)—Expressed in *E. coli* at ~1% (*w*/*w*) of the Cell Soluble Protein.**											
F140I		solubility	^c^ WT	^c^ ↑ **46%**	↓ **48%**	-	↑ **288%**	-	↓ 9%	MM1	[27]
			↑ **4.9-fold**	↓ 12%	-	↓ **16%**	-	↑ **6%**	WT	RDE2	[22]
V189I		^d^ solubility	WT	↑ 8%	↓ **35%**	-	↑ **71%**	-	↓ 12%	MM1	[27]
			↑ **4.8-fold**	↓ 16%	-	↓ **26%**	-	↑ **14%**	↓ 15%	RDE2	[22]
			-	↓ 25%	↓ **44%**	↓ **41%**	↑ **32%**	↑ **8%**	-	*R. capsulatus*	[28]
M262T		^d^ solubility	↑ **6-fold**	↑ **13%**	↓ **13%**	-	↑ **29%**	-	WT	RDE(K12)	[29]
			↑ **4-fold**	-	-	-	-	-	-	MM1	[22]
			-	WT	↓ 23%	↓ 15%	↑ 23%	↑ 18%	-	*R. capsulatus*	[28]
F345I		solubility	↑ **8-fold**	↓ 14%	↑ 4%	↑ 10%	↓ 17%	↓ 22%	↓ 13%	MM1	[30]
			↑ **6-fold**	↓ 15%	↑ 10%	-	↓ 21%	-	↓7%	MM1	[27]
			↑ **11.1-fold**	↓ 16%	-	↑ 13%	-	↓ 28%	↓ 8%	RDE2	[22]
F345L		solubility	↑ **7-fold**	↓ 17%	↑ 22%	-	↓ 32%	-	↓4%	MM1	[30]
F345V		solubility	-	↓ 25%	↑ 38%	-	↓ 45%	-	↓ 45%	*R. capsulatus*	[31]
***Te*-Rubisco (*Thermosynechococcus elongatus* BP-1)—Expressed in *E. coli* at ~6% (*w*/*w*) of the Soluble Protein.**											
F345I		solubility	↑ **2.2-fold**	↓ 27%	-	WT	-	↓ 24%	↓ 10%	RDE2	[22]
P415A		solubility	↑ **1.6-fold**	↑ **5%**	-	WT	-	WT	WT	RDE2	[22]
P415A	F345I	solubility/kinetic	↑ **2.7-fold**	↓ 27%	-	WT	-	↓ 24%	↓ 8%	RDE2	[22]
P415A	V98M^S^	solubility/kinetic	↑ **3.1-fold**	↑ **44%**	-	↓ 13%	-	↑ 44%	↑ 6%	RDE2	[22]
P415A	A48V^S^	kinetic	↑ **1.5-fold**	↑ **33%**	-	WT	-	↑ 35%	↓ 10%	RDE2	[22]
P415A	H37L^S^	kinetic	↑ **1.7-fold**	↑ **27%**	-	WT	-	↑ 28%	↓ 7%	RDE2	[22]
P415A	Y36N^S^/G112D^S^	kinetic	↑ **1.5-fold**	↑ **41%**	-	↓ 12%	-	↑ 60%	↓ 10%	RDE2	[22]
P415A	L74M/D397N	solubility/kinetic	↑ **1.9-fold**	↑ **24%**	-	↓ 14%	-	↑ 45%	= WT	RDE2	[22]

Amino acid numbering relative to plant RbcL (Figure A1). Data summarized from the Rubisco content and kinetic listings in Table A1. ^a^ The governing biochemical feature that improves RDE fitness (i.e., cell growth rate). ^b^ A directed evolution screen is used to isolate the mutant (see Figure 1). The *R. capsulatus* screen is a low transformation frequency system that uses a Rubisco null mutant [31]. ^c^ Underestimates of Rubisco content likely explain the erroneous carboxylation rate and efficiency improvements. ^d^ Improvements in carboxylase properties may contribute to the improved fitness that is primarily imparted by the >4-fold increases in Rubisco biogenesis (solubility). Enhanced carboxylase activities are shown in bold-type. Arrows indicate whether the kinetic value is higher (**↑**) or lower (↓).

**Table 2 ijms-20-05019-t002:** Comparative *E. coli* biogenesis (solubility) and catalysis (at 25 °C) of wild-type, YZ1 and their *Rs*Rubisco chimers.

*Rs*Rubisco Isoform	RbcL Residue Number				Rubisco	*K_C_*	*k_cat_^C^*	*K_cat_^C^/K_C_*	*S_C/O_*	*K_O_*	*k_cat_^O^*	*k_cat_^O^/K_O_*
	11	83	252	345	^a^ (% CSP)	(μM)	(s^−1^)	(mM.s^−1^)	mol·mol^−1^	(μM)	(s^−1^)	(mM·s^−1^)
WT	Val	Lys	Arg	Tyr	4.9 ± 0.4 ^a^	60 ± 1 ^a^	3.7 ± 0.2 ^a^	60 ± 1	58 ± 1 ^a^	1724 ± 280	1.8	1.0
YZ1	Ile	Gln	Leu	Phe	2.9 ± 0.1 ^b^	66 ± 1 ^a^	**4.7 ± 0.1** ^c^	**70 ± 1**	57 ± 1 ^a^	1296 ± 207	1.6	1.3
YZ1a	Val	Gln	Leu	Phe	3.3 ± 0.4 ^b^	66 ± 1 ^a^	**4.5 ± 0.1** ^c^	**68 ± 2**	-	-	-	-
YZ1b	Ile	Gln	Leu	Tyr	3.0 ± 0.1 ^b^	61 ± 1 ^a^	3.8 ± 0.1 ^a^	63 ± 1	-	-	-	-
YZ1c	Val	Gln	Leu	Tyr	3.6 ± 0.a ^b^	63 ± 1 ^a^	3.9 ± 0.1 ^ab^	63 ± 1	-	-	-	-
YZ1d	Ile	Gln	Arg	Phe	4.6 ± 0.1 ^a^	62 ± 1 ^a^	**4.6 ± 0.1** ^c^	**76 ± 1**	-	-	-	-
YZ1e	Ile	Lys	Leu	Phe	4.5 ± 0.3 ^a^	63 ± 1 ^a^	**4.3 ± 0.1** ^bc^	**70 ± 3**	-	-	-	-
YZ1f	Val	Gln	Arg	Tyr	5.1 ± 0.4 ^a^	61 ± 2 ^a^	3.9 ± 0.2 ^ab^	65 ± 1	57 ± 1 ^a^	1220 ± 199	1.4	1.1
YZ1g	Val	Lys	Leu	Tyr	5.3 ± 0.2 ^a^	62 ± 4 ^a^	3.5 ± 0.1 ^a^	57 ± 3	-	-	-	-
YZ1h	Ile	Lys	Arg	Phe	4.6 ± 0.1 ^a^	61 ± 1 ^a^	**4.4 ± 0.1** ^a^	**73 ± 1**	-	-	-	-
YZ1i	Val	Lys	Arg	Phe	4.7 ± 0.2 ^a^	64 ± 1 ^a^	**4.6 ± 0.1** ^a^	**71 ± 2**	59 ± 1 ^a^	1551 ± 293	1.9	1.2

Values in bold represent significant improvements in carboxylase activity. CSP (percentage (*w*/*w*) of the cell soluble protein), *k_cat_^C^* and *K_C_* data are the mean ± SE of *N* = 5 (and for S_C/O_
*N* = 2) biological samples, each assayed in duplicate. Letters show the ranking of the means using a post hoc Tukey test (different letters indicate statistical differences at the 5% level, *p* < 0.05) following a one-way ANOVA comparison relative to WT. *k_cat_^O^*, maximal oxygenation rate, calculated from *S_C/O_* = (*k_cat_^C^ /K*_C_*)*/(*k_cat_^C^/K*_O_). Mutated residues are shaded black.

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
