# Peer review of "Directed Evolution of an Improved Rubisco; In Vitro Analyses to Decipher Fact from Fiction"

_ijms, 2019, doi:10.3390/ijms20205019_

Round 1
Reviewer 1 Report
The authors described an analytical pipeline for quantifying Rubisco content and kinetics that averts misinterpretation of directed evolution outcomes. This analytical pipeline is suitable for identifying false positives from directed evolution experiments targeted on the Rubisco. The paper can be accepted for publication.
Author Response
We thank the reviewers for their favourable assessment of our manuscript
Reviewer 2 Report
1, Sometimes RsRubisco is expressed as Rs-Rubisco.
(Sometimes SeRubisco is written Se-Rubisco.)
2, Expression on Figure 4 is described in the section of "Discussion", at the first time. Could you consider the order of Figures or the position of Figure 4 on the paper.
Author Response
1, Sometimes RsRubisco is expressed as Rs-Rubisco.
(Sometimes SeRubisco is written Se-Rubisco.)
During our revisions of the manuscript we have standardised our nomenclature to RsRubisco and SeRubisco in the text, tables and Figures
2, Expression on Figure 4 is described in the section of "Discussion", at the first time. Could you consider the order of Figures or the position of Figure 4 on the paper.
Reference to Figure 4 is mentioned in the results section on line 224 (line 219 previously)
Reviewer 3 Report
The manuscript by Zhou and Whitney describes an approach based on directed evolution and biochemical analyses aimed at generating and identifying/analysing an improved RubisCO enzyme, respectively. This is an important topic and the manuscript provides a rationale design approach to identify enzyme mutants with better features in terms of both kinetics and protein solubility compared to the wild-type form. Although the manuscript contains interesting data, some sections are confusing and data presentation suffers quality and not fully supported by figures.
In order to improve the quality and clarity of the manuscript, a general reorganization of the results is strongly recommended. This will allow to better guide the readers into the experimental pipeline that appears to have a strong impact on the comprehension of RubisCO catalysis and biogenesis.
In Fig. 1, the protein amount loaded per lane is shown (see Fig. 1). However, if each lane was loaded with 4.0 micrograms there is a substantial problem in protein quantification before loading. Moreover, the protein amount per lane is not equal as stated.
Regarding Fig. 2, the authors did not provide any comments on recombinant red-type RsRubisCO. This must be added here even if is commented lately in the text. Paragraph 2.1 of results contains statements already mentioned in the introduction.
The result section 2.2, no figures or table support authors’ claims and it is not clear if these sentences refer to previous studies.
In Fig.6, the lane corresponding to the empty vector contains much less protein amount and could not be considered as a reliable control.
In general, the conclusion section should appear just after discussion and not at the end of the manuscript. However, I am not aware of journal guidelines but I suggest the authors to check this point.
Is there any correspondence for Val11 in tobacco RubisCO?
Figure A1 and A2 should be clearly indicated as Appendix material
A careful re-reading is mandatory to fix unclear sentences and typing errors
Author Response
In order to improve the quality and clarity of the manuscript, a general reorganization of the results is strongly recommended. This will allow to better guide the readers into the experimental pipeline that appears to have a strong impact on the comprehension of RubisCO catalysis and biogenesis.
In the absence of specific suggestions we have been unable to identify how the results section could be re-organised to improve clarity. We note that no similar concerns were raised by the other reviewers.
In Fig. 2, the protein amount loaded per lane is shown (see Fig. 2). However, if each lane was loaded with 4.0 micrograms there is a substantial problem in protein quantification before loading. Moreover, the protein amount per lane is not equal as stated.
We have modified the labelling and legend to Figure 2 to improve clarity around the protein loadings in the gels and blots. Sample loading in the Coomassie stained gel is normalised with regard to soluble protein assays. The same volume of protein from the Lysate sample is then loaded (which contains 4 ug of soluble protein as well as insoluble membrane and inclusion body proteins). Less amounts of the samples are used in the immune-blot assays. We hope the protein loading information in the new figure is more easily understood.
Regarding Fig. 2, the authors did not provide any comments on recombinant red-type RsRubisCO. This must be added here even if is commented lately in the text.
A good suggestion. We have now added in the sentence “By comparison the assembly requirements of RsRubisco are more readily met in E .coli with a greater proportion of soluble RsRbcL produced and assembled into RsL8S8 complexes (Figure 2c).” (Line 153-155)
Paragraph 2.1 of results contains statements already mentioned in the introduction.
Unlike the information provided in the introduction, the paragraph in question specifies for the first time specifies the errors that exits in the literature in regard to mutations that improve solubility being misinterpreted as catalytic enhancing mutations. This paragraph functions as the place setter for why analytical rigour is needed when undertaking and interpreting Rubisco directed evolution studies.
The result section 2.2, no figures or table support authors’ claims and it is not clear if these sentences refer to previous studies.
A good point, we had forgotten to cite references where we have used the analytical pipeline in out directed evolution studies. They are now cited at the beginning of the paragraph (line 179)
In Fig.6, the lane corresponding to the empty vector contains much less protein amount and could not be considered as a reliable control.
A valid criticism. After every series of kinetic assays we separate the soluble protein samples by native PAGE to confirm the accuracy of the corresponding Rubisco quantification by 14C-CABP binding. The native PAGE shown originally was chosen as it showed the best protein separation, however, as noted in the original legend, only 2ug of the control sample was accidently loaded, not 8ug. In the revised version we have now provide a picture of a native PAGE from another experiment – although less soluble protein (4ug) from each cell lines is loaded, but the outcome is still the same with regard to showing the differing amounts of assembled RsRubisco produced in each RsYZ1 chimer.
In general, the conclusion section should appear just after discussion and not at the end of the manuscript. However, I am not aware of journal guidelines but I suggest the authors to check this point.
The conclusion is located in the position stipulated in the template provided by the journal.
Is there any correspondence for Val11 in tobacco RubisCO?
We are not aware of any published information about the influence of mutations in this N-terminal region having any influence on plant Rubisco catalysis. From some of our unpublished work we know that our N-terminal modifications introduced onto the RsRbcL do not alter KcatC, Kc or Sc/o. This information forms part of a future publication that is awaiting a comparative measure of Ko.
Figure A1 and A2 should be clearly indicated as Appendix material
The location and headings were again defined by the template provided by the journal.
A careful re-reading is mandatory to fix unclear sentences and typing errors.
We have endeavoured to fix the grammar and multiple typing errors in the revised version.
Reviewer 4 Report
The manuscript by Zhou et al describe a new pipeline for directed evolution of Rubisco, a CO2-converting photosynthetic enzyme. The author describe the reason for the need of an improved pipeline by highlighting the limits of the currently used ones. In particular, they note the frequency of false positive and the inability to distinguish mutation that improve the solubility/stability of its components vs ones that improve the catalytic activity. They next describe in details their improved pipeline with numerous analytical experiments and their choices of bacterial strain (JM109(DE3)) and source of Rubisco (R Sphaeroides). They present the results of their screen and characterize the selected mutant (4 mutations) together with chimera that incorporate different combination of these mutations (from 1 to 3). Analysis of the various mutants allows them to confirm the catalytic improvement of the selected clone and better understand the role of these mutations.
The article is clearly written and very well presented. The conclusions are supported by the results.
Minor comments:
-minor typos to be corrected
-could the authors explain the meaning of the red color in table 1
Author Response
Minor typos to be corrected
We have endeavoured to fix the grammar and multiple typing errors in the revised version.
Could the authors explain the meaning of the red color in table 1
This color was accidently left in the original version and has now been removed.
Round 2
Reviewer 3 Report
The authors nicely addressed all my concerns and I consider the present version of the manuscript worth of publication in the IJMS journal